# FPGA-Based Acceleration on Additive Manufacturing Defects Inspection

**DOI:** 10.3390/s21062123

**Published:** 2021-03-18

**Authors:** Yawen Luo, Yuhua Chen

**Affiliations:** Department of Electrical and Computer Engineering, University of Houston, Houston, TX 77204, USA; yuhuachen@uh.edu

**Keywords:** FPGA, additive manufacturing, defects inspection, convolutional neural network, binarized neural network, selective search

## Abstract

Additive manufacturing (AM) has gained increasing attention over the past years due to its fast prototype, easier modification, and possibility for complex internal texture devices when compared to traditional manufacture processing. However, potential internal defects are occurring during AM processes, and it requires real-time inspections to minimize the costs by either aborting the processing or repairing the defect. In order to perform the defects inspection, first the defects database NEU-DET is used for training. Then, a convolution neural network (CNN) is applied to perform defects classification. For real-time purposes, Field Programmable Gate Arrays (FPGAs) are utilized for acceleration. A binarized neural network (BNN) is proposed to best fit the FPGA bit operations. Finally, for the image labeled with defects, the selective search and non-maximum algorithms are implemented to help locate the coordinates of defects. Experiments show that the BNN model on NEU-DET can achieve 97.9% accuracy in identifying whether the image is defective or defect-free. As for the image classification speed, the FPGA-based BNN module can process one image within 0.5 s. The BNN design is modularized and can be duplicated in parallel to fully utilize logic gates and memory resources in FPGAs. It is clear that the proposed FPGA-based BNN can perform real-time defects inspection with high accuracy and it can easily scale up to larger FPGA implementations.

## 1. Introduction

Additive manufacturing (AM), involving components production layer by layer, allows users to create a product directly from a 3D computer model, while the conventional subtractive processing is the result of removal of material, formative processes, and joining processes [1,2,3]. The nature of adding materials from the bottom up makes AM a perfect solution for manufacturing dedicated, complex components. In addition, AM yields faster prototypes and is convenient for a design modification. Moreover, AM makes it possible for object components to come with various colors, textures, and materials. When it comes to manufacturing complex lattice structures, AM is the only solution [4]. Although AM is gaining tremendous attention due to these advantages, it is also more susceptible to internal defects. Therefore, defects inspection is a major concern for AM and it needs to be resolved in real-time automatically.

Defects inspection for traditional subtractive processing is usually performed after the entire object is produced, which typically only guarantees the surface quality. Meanwhile, in some factories, the product inspection is conducted by human vision. It is labor-intensive and the accuracy is limited. Apparently, those methods are not suitable in AM. Chua et al. [5] reviewed current process monitoring/control systems for metal AM and proposed a comprehensive real-time inspection method and a closed-loop monitoring system to improve the quality of AM printed parts. To ensure the internal quality while improving the efficiency of AM product inspection, deep learning is commonly applied in both industry and academia [6,7,8]. Typically, the deep learning-based defects inspection involves three steps: image acquisition, image classification, and defects localization. Generally, a sophisticated high-speed, high-resolution camera is deployed to capture object surface images during AM processing.

Zhu et al. [9] applied the convolution neural network (CNN) for feature extraction and tried different classifiers: random forest, SVM, and soft-max to achieve 98.75% accuracy on their own dataset. The experimental environment was a 64-bit Windows 10 system with i5 CPU, 8 GB memory, and 2.30 GHz basic frequency. However, their method was time-consuming and could not be applied to real-time industrial applications. Huang et al. [10] introduced a novel compact CNN model with low computation and memory cost and achieved 100% accuracy and 29 ms inference time on the NEU dataset with Intel i3-4010U CPU. Cui et al. [11] presented a CNN model which achieved an accuracy of 92.1% within 8.01 ms. The experiments were conducted with AMD Ryzen 52600 processor and Nvidia 1070 GPU. He et al. [12] proposed a defect detection network (DDN) and set up the NEU-DET dataset. Experiments showed that their DNN obtained 99.67% accuracy for defect classification and 82.3 mAP for defect detection. However, the authors did not present image inference time and hardware configurations. Above researches, while reaching the defects classification goal with high accuracy and promising inference time, were all CPU-based and GPUs were used for acceleration purposes.

Field Programmable Gate Arrays (FPGAs) have high throughput and can inherently provide deterministic low latency for real-time applications [13,14]. Compared to GPUs, FPGAs are more power-efficient and designers can build the neural network from ground up and tailor the FPGA to best fit the model. Moreover, FPGAs allow camera sensor fusion and captured images can be handled by the FPGA directly with minimal latency. ARM processors are generally great for the low-power environment but are typically slower when compared to Intel CPUs [15]. However, modern FPGAs have embedded ARM processors along with custom logic on the same chip, striking a balance between performance and flexibility. In fact, the AM machine itself is an embedded system. It is desirable that the solution can be incorporated as part of the AM machine, so it can perform the defect inspection automatically. This will make the AM machine with defects inspection a standalone system, which is more attractive from the commercial perspective.

Mujawar et al. [16] proposed a 3-layer CNN architecture targeting at written digits recognition application on the MNIST dataset and implemented it in Artix-7 FPGAs. The authors also optimized the architecture by using loop-level parallel processing. Simulation results showed that the FPGA version can achieve 90% accuracy and 58 μs inference time. However, image sizes (28 × 28) in the MNIST dataset are much smaller than the NEU-DET defects dataset (200 × 200). Nakahara et al. [17] implemented a multi-scale sliding window-based object detector on an FPGA. They trained the VGG11-based binarized convolutional neural network (BCNN). Their detector could classify the image and locate the object. The implementation was on a Xilinx Zynq UltraScale+ MPSoC zcu102 evaluation board. The researchers did a comparison between their proposed ARM-FPGA detector and the well-known YOLOv2 CPU-GPU version. Their proposed FPGA-based solution could achieve 82.20% accuracy, 28 ms inference time, and consume only 2.5 W. In comparison, the YOLOv2 GPU accelerated version achieved 28.37% accuracy, 20 ms inference time and 250 W.

In the AM process, once any type of defect is identified, AM machine should pause the production immediately and try to further inspect or fix the defect area based on specific defect types. This is because it would be wasting materials and time on a defective product unless the flaws are determined to be acceptable or are repaired. Therefore, locating the defects should be executed right after defects classification. Nakahara et al. [17] used the sliding window on multi-scale images to produce a set of sub-images. After the sub-images were resized to the original size, they were wrapped together and sent to the FPGA-based classifier. For any sub-image labeled with defects, its corresponding coordinates in the original image will depict the defects area. However, the proposed multi-scale sliding window is time-consuming and many defect-free sub-images are generated unnecessarily. In order to produce the potential sub-images with defects more efficiently, region proposal algorithms can be deployed. Van de Sande et al. [18] adapted segmentation as a selective search by reconsidering segmentation and proposed to generate many approximate locations over precise object delineations. Pont-Tuset et al. [19] developed a fast normalized cuts algorithm, and then proposed a high-performance hierarchical segmenter that made effective use of multiscale information. Finally, the authors proposed a grouping strategy that combined multiscale regions into highly-accurate object regions by efficiently exploring their combinatorial space. Taghizadeh et al. [20] proposed an efficient algorithm to generate a limited number of regions for resolving visual recognition problems.

In this paper, we propose an FPGA-based embedded design to perform AM defects identification and classification in real time. Although defects inspection exists, most of the methods were implemented on CPU-GPU and cannot be directly integrated into AM machines. Our method is an embedded FPGA solution and can be readily adopted. The CNN architecture is first presented to achieve high classification accuracy. Next, binarization is applied to CNN parameters to obtain the binarized neural network (BNN) architecture. Then, the BNN is implemented in the FPGA for acceleration purposes. Finally, the FPGA-based BNN model is used for AM defects inspection. Once defects are detected, the selective search algorithm and non-maximum suppression algorithm are applied to locate the defect regions.

The outline of this paper is organized as follows. Section 2 shows research methods. More specifically, defect image acquisition, neural network architecture, FPGA implementation, and defects localization are described. Section 3 presents the results and evaluates the performance. The Section 4 draws the conclusion.

## 2. Research Methods

### 2.1. Defect Image Acquisition

Materials used in AM can be polymers and metals [21]. Due to differences in material property, defect features in metal AM and polymer AM have remarkable disparities. In our paper, we studied the metal additive manufacturing. Acquiring the defect image dataset is usually the barrier that stops researchers from training their neural network since obtaining their own dataset from experiments consumes lots of time and materials [22].

#### 2.1.1. NEU-DET

K. Song and his team’s efforts on the Northeast University surface defect database (NEU-DET) [23] helped alleviate the above situation in the defects detection field. It includes six kinds of typical surface defects of the hot-rolled steel strip. In our work, we took four types of defects from NEU-DET, namely rolled-in scale, patches, inclusion, and scratches for model training. There are 300 samples for each type of defect, and each image has 200 × 200 pixels, where the pixel size is 1.975∼2.7 mm (depending on images). The 200 × 200 image size is typical for training CNNs. Resolutions beyond that, however, will tremendously increase the size of the neural networks and slow down the training and prediction speed. When the image resolution is higher, a common practice is to split the image into sub-blocks and use the sub-images for training the CNN model to perform defect inspection in metal AM [11]. The 200 × 200 image size used in our paper is comparable to literature with CPU-GPU approaches, and is considered the state-of-the-art comparing to other FPGA-based implementations (e.g., 28 × 28 in [16], 32 × 32 and 48 × 48 in [17]). Zhu et al. [9] applied images with the size of 120 × 80 pixels to train the CNN for classification of weld surface defects. Our work is applicable to defective images at different scales as long as the images are segmented into sub-images.

#### 2.1.2. Preprocessing

The 200 × 200 image size of the NEU-DET dataset is sufficiently large to train our neural networks. For each image in the NEU-DET dataset, we extracted a defect-free sub-region, resized it to 200 × 200 and labeled it as “defect-free”. In addition, according to the defect annotations, for every image in each defect type, we extracted the sub-regions, resized them into 200 × 200, and labeled them with the same defect type. This ensured that the trained neural network could be also used for locating the defects. The modified dataset was split into training and testing sets at the ratio of 7:3. For every 10 random images in the dataset, seven images were assigned to the training set, and the other three were assigned to the testing set.

### 2.2. Neural Network Architecture

CNN models are ubiquitous in image data processing. It is one of the state-of-the-art computer vision techniques for image classification [24,25]. In general, the number of parameters in neural networks grows tremendously with the number of layers, which makes the model training computationally intensive. CNNs’ convolutional operations use information from adjacent pixels to effectively downsample the image. It can successfully extract the image features while reducing the number of neural network parameters. Therefore, in this paper, we adopt the CNN model due to its efficiency and high accuracy. However, considering the fact that we will implement it on FPGAs because of the energy efficiency and embedded nature of the design, we binarize the CNN weight parameters. With binarized weight values, we can replace the multiplication with XNOR bit operations, which tremendously speeds up the inference time in the FPGA [26,27]. Therefore, a BNN architecture as shown in Figure 1 is proposed in our studies.

Our BNN model is elaborated as follows. The 200 × 200 pixels gray-scale image is first processed by:(1)f(x)=(x−127.5)/128.
The *x* is a pixel represented by 8-bit integer, which ranges 0 to 255. The constant 127.5 is the mean of range boundaries 0 and 255. The constant 128 is a scaling factor. After the pre-processing, the pixel values are constrained within (−1, 1). Then the matrices of real numbers are fed into the first convolutional layer with a filter size of 3 × 3 for feature extraction. In the convolutional layer, the “use_bias” parameter is set to false. The max pooling layer is used for extracting the maximum value in a sub-region of the feature map. Our max pooling layer is of 2 × 2 in size and the stride is set to 2. The output of max pooling is without padding. Batch normalization layer is applied to mitigate the effect of unstable gradients. It standardizes and normalizes the input values by scaling and shifting operations. In our batch normalization layer, momentum is set as 0.999, epsilon is 0.001, and scale is false, respectively. The momentum is a parameter used for calculating the population average during inference. The epsilon is a small float number added to variance to avoid dividing by zero. The binarization layer works as an activation function as shown in the following equation:(2)$$q\left(x\right)=\left\{\begin{array}{cc} -1, & x<0\\ 1, & x\geq 0, \end{array}\right.$$
where *x* is the output from the batch normalization layer, which is a real number. The q(x) is a 1-bit output of the binarization layer. After passing three additional convolutional layers as well as corresponding max pooling, normalization, and binarization layers, we can successfully extract image features.

Once the features are extracted, the next step is classification. The classification module takes 25 × 10 × 10 feature maps and flattened them as a 2500 feature vector. Then, two dense layers are deployed to densify the 2500 feature vector to the dimensional of 5. Finally, a soft-max layer, as depicted in Equation (Equation 3), is applied to generate the output. In our BNN design, *M* is the number of classes (number of defect types plus one defect-free type). $Z_{i}$ represents the *i*th real number from the BNN activation outputs. $\delta (z_{i})$ is the probability of the *i*th class. The final outputs are 5 real numbers representing the probabilities of defect types, namely, defect-free, rolled-in scale, patches, inclusion, and scratches.
(3)$$\delta \left(z_{i}\right)=\frac{e^{z_{i}}}{\sum_{j=1}^{M}e^{z_{j}}}\;for\;i=1,\ldots ,M\;and\;z=\left(z_{1},\ldots ,z_{M}\right)\in R^{M}.$$

In our BNN model, the cross-entropy loss function, as shown in Equation (Equation 4), is applied for measuring the differences between the predicted class and the corresponding labeled class:(4)$$L\left(w\right)=\sum_{m=1}^{M}\sum_{c=1}^{C}y_{mc}log\left(P\left(y_{m}=c|X_{m}\right)\right),$$
where *M* represents the number of classes, $y_{m}$ is the predicted class, and *c* is the labeled class. The images with label *m* is denoted as $X_{m}$. The training process of our model is targeting at minimizing the loss function by adjusting the weight parameters. Our BNN training is performed on an Intel i5-7400 CPU @ 3.000 GHz. We used python scripts with the open-source library of larq [28] and keras [29]. The TensorFlow backend engine was employed, and the NVIDIA GeForce GTX 1060 3GB was applied for acceleration.

### 2.3. FPGA Implementation

After the BNN model is trained on the CPU, it can be used for inference. In order to make the AM machine an independent and complete system, we present the inference process as an embedded design. The BNN is implemented in the system on a chip (SoC) [30], which is the Terasic SoC kit development board as shown in Figure 2. This board contains an FPGA embedded with an ARM processor, as well as other components such as memory, connectors, sensors, and display. The FPGA device is Intel Cyclone V. The hard processor embedded in the FPGA is a Dual-Core ARM Cortex-A9 MPCore Processor. This board is affordable and is currently available in common electronic components distributors such as Digi-Key Electronics. It can also be ordered directly from Terasic’s official website. Due to its low price, it is usually used for education purposes and small system prototypes.

The FPGA acceleration architecture is described in Figure 3. The image prepossessing, convolutional layer 1, and the last activation layer are realized in the ARM. As a result that these layers involve real number arithmetics, it is easier to be implemented in the ARM instead of the FPGA custom logics. The outputs of convolutional layer 1 are sent to the FPGA through the AXI bus using the Xillibus IP core [31]. After FPGA processing, the output of dense layer 2 is then transferred back to the ARM for the activation layer.

In our BNN model, only the convolutional and dense layer weights are binarized, while the parameters of the batch normalization layer are still real numbers. Figure 4 shows the details. Although the drawing only shows part of the entire BNN model, it is clear that the values become real numbers right after passing through the normalization layer, which indicates that the weights of the normalization layer are real numbers. In general, a real number is represented as a floating point number in computers. In order to avoid any floating point operations in the FPGA, which are resource hungry and timing consuming, we use the fixed-point notation [32]. The fixed-point number representation is a real data type for a number that has a fixed number of digits after the radix point. A value of a fixed-point data type is essentially an integer that is scaled by a specific factor determined by the type. In our design, the scaling factor is 1024, which means that 10 bits are needed for representing the original real number’s fraction part. According to the maximum value of our batch normalization parameters, we use 18 bits to represent the integer part of the original real number. As a result, the real number is stored as a 28-bit fixed-point number inside the FPGA.

Our FPGA design is written in Verilog hardware description language (HDL), which is used to model electronic systems. Verilog is commonly used in the design and verification of digital circuits at the register-transfer level. Different from software design using c/c++, hardware design using HDLs need to take care of the timing. HDLs use principal digital blocks such as flip flops, registers, latches, and memory to describe digital circuits. It has to handle address and clock signals and take the gates/propagation delay into account. In our FPGA design, we use finite state machines (FSMs) to help implement the different convolutional layers in the BNN model. The FSM is a computation model that can guarantee the logic sequence of a function and is usually applied in synchronous designs.

In the FPGA design, the convolution operation can be taken as a set of matrix multiplications. Those intermediate results are stored in the random-access memory (RAM). RAM access time is typically 2 clock cycles. Our convolutional layer filter size is 3 × 3. Before every matrix multiplication, it needs to repeat 9 times the image data access to RAM, which could be time-consuming. Instead, we define a shift register as shown in Figure 5 to access all 9 data at once [17], which speeds up the process dramatically. This will be even more efficient if a larger filter size is deployed (e.g., 5 × 5 or 7 × 7).

During the inference process, all the layer weights are stored in the read-only-memory (ROM), which is a type of non-volatile memory used in embedded systems. Data stored in ROM cannot be modified after it is programmed. This makes it perfect to store the trained neural network weights/parameters. In the FPGA design, we need first to prepare the memory initialization files (with the extension .mif) that specify the ROM initial contents.

### 2.4. Defects Localization

After any image is classified as one of the four defect types: rolled-in scale, patches, scratches, and inclusion, the AM machine should stop printing the object and further inspect the defect areas. To make our AM defect inspection more functionally powerful, we provide the defect locations based on the existing BNN model. The image identified with defects should be retrieved for further processing. Given its size 200 × 200, we use a 4-tuple $<x_{min},x_{max},y_{min},y_{max}>$ that contains four coordinates to constrain the defect area. The $x_{min}$, $x_{max}$, $y_{min}$, and $y_{max}$ are defect area’s left boundary, right boundary, up boundary, and bottom boundary, respectively, with the range from 1 to 200. We need to find a 4-tuple value for every defect within that image. Our approach is to suggest as many potential defect sub-regions as possible, where each sub-region is denoted by a 4-tuple. Then those sub-regions will be resized to 200 × 200 pixels and fed into the BNN model. If the output of a sub-region is labeled as defective, its corresponding 4-tuple is marked as one of the defect areas, which can be used for defects repair or displaying on the screen to alert the specialist.

#### 2.4.1. Sub-Region Generation

The objective of sub-region generation is to identify the possible locations of the defects in the image and detect more defects while fewer sub-regions are generated. Selective search algorithm is typically used to efficiently propose multiple sub-regions [19,33]. It starts by over-segmenting the image. Then, it recursively combines smaller similar regions into larger ones. Selective search takes four types of similarities into account, namely, color similarity, texture similarity, size similarity, and shape similarity. When the number of regions, which decreases as the recursive combination continues, reaches a pre-defined threshold, the rest regions are treated as candidate sub-regions.

#### 2.4.2. Non-Maximum Suppression

However, the previously mentioned selective search algorithm will generate many overlapping regions. If one defect object is included within several regions, the BNN would output multiple redundant 4-tuples. More illustrative details are shown in Figure 6. To eliminate those overlapping regions, we adopted the non-maximum suppression [34]. The non-maximum suppression method takes a list of bounding boxes and corresponding defect type probabilities as inputs, where the boxes are described by 4-tuples, and selects a single bounding box out of many overlapping boxes. Non-maximum suppression works as follows:Discard boxes that are below a given probability threshold $T_{1}$.Pick the box with the highest probability as the output.Discard remaining boxes where $IoU\geq T_{2}$ with the previous output box. IoU is the intersection over Union. $T_{2}$ is a given threshold, and typically set to 0.5.Repeat step 2 and 3 until no boxes left.

## 3. Results Evaluation

### 3.1. Evaluation of BNN Architecture

The testing set images are used for evaluating our BNN model. For every image fed into our BNN model, five outputs are generated to represent the probabilities of each class. The class with the largest probability is our BNN model’s prediction. The accuracy is calculated by the number of correct predictions over the total number of predictions. Our BNN model can achieve 97.9% accuracy on either defective or defect-free classification. This is achieved by treating all four defect types: rolled-in scale, inclusion, scratches, and patches as defective only. We also evaluated the classification accuracy on specific defect types for repair purposes. Due to the cross-class similarity, various defect types do not have apparent differences and different types of defects overlap within some images. Even for the same defect class, training defect images are acquired under different illuminations and there are changes in materials. Our defect type classification accuracy is 88.07%. Relative BNN model’s accuracy and loss plots are shown in Figure 7. We believe that the 88.07% accuracy on defect type classification is acceptable for our BNN model. In AM defects inspection, the most important goal is to distinguish whether the object has defects or not, in which case the exact defect type is not of high priority.

### 3.2. Evaluation of FPGA Implementation

Our embedded design of the BNN model is used for inference and a single BNN implementation can inspect one image within 0.5 s. The BNN design is modularized and can be replicated to fully utilize logic gates and memory resources of FPGAs. The FPGA design compilation results are shown in Figure 8. The logic gates utilization is only 12% and block memory utilization is 22%. Each BNN module utilizes 5 K ALMs and 1.25 Mb block memory. Our design can provide real-time inspection capability since AM process is a mechanical process and is relatively slow. The FPGA device on the Terasic SoC kit development board belongs to Intel Cyclone V series, which is a lower-end family of Intel SoC FPGAs. Other SoC FPGA devices (from Intel or Xilinx) can also be used for our BNN model implementation without many modifications. Therefore, our design can easily scale up and inspect more images simultaneously by switching to a more powerful FPGA chip. For example, Intel FPGA Stratix 10 GX10M can generate 246 BNN modules, which is capable of processing one image every 2 ms. Table 1 shows three Intel FPGA families with different logic gates and block memory sizes.

### 3.3. Evaluation of Defects Inspection Results

For every image fed into our BNN inference model, we have five real number outputs. The largest value means the highest probability and its corresponding class is the expected result. For any image identified as defect-free, no further actions are needed. The AM machine just keeps printing the object, and the camera continues to send images to the BNN module. Once any defect is spotted, the AM machine stops printing. Our ARM processor will execute the selective search algorithm to propose multi-regions. After resizing those regions, the ARM will feed them into the BNN and record their corresponding coordinates in the original image. In addition, non-maximum suppression algorithm is adapted to eliminate overlapping areas. Finally, defect areas will be provided and highlighted with boxes. Some sample defect results are shown in Figure 9.

## 4. Conclusions

In this paper, we have proposed an FPGA-based real-time embedded system for AM defects inspection. The BNN model was presented and implemented on FPGA. Including identifying defect types, our design can also provide defect area coordinates for any follow-up repairs or specialist’s re-verification by utilizing selective search and non-maximum suppression algorithms. The NEU-DET dataset is used for training our BNN model. It has achieved 97.9% accuracy in identifying whether captured object image has defects or not. By applying FPGA acceleration, our BNN model can process a single gray-scale image within 0.5 s. Considering the AM object production speed, the results can guarantee real-time quality inspection in the manufacturing process. In addition, our FPGA-based design can easily scale up by generating more BNN modules in a more powerful FPGA. It is also promising for other applications with higher processing speed requirements. 

## Figures and Tables

**Figure 1 sensors-21-02123-f001:**
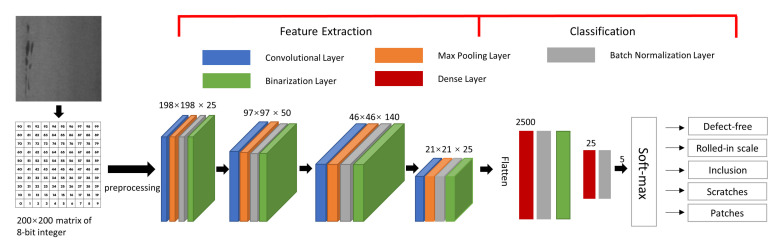
Binarized neural network architecture.

**Figure 2 sensors-21-02123-f002:**
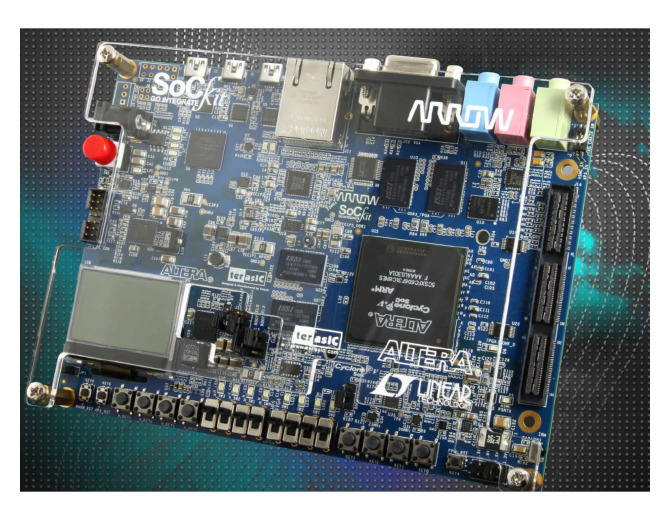
Terasic system on a chip (SoC) kit development board.

**Figure 3 sensors-21-02123-f003:**
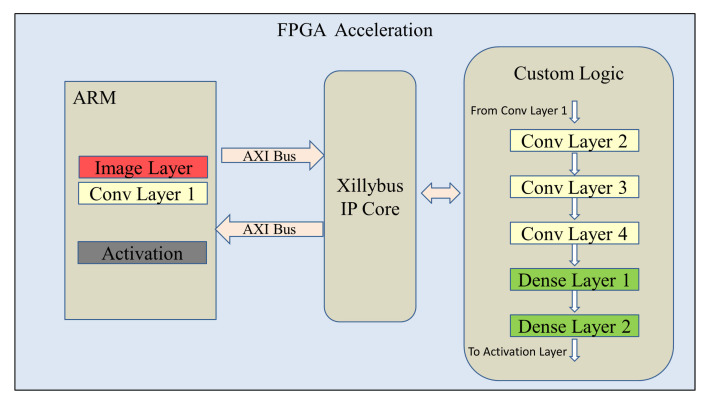
Field Programmable Gate Array (FPGA) acceleration architecture.

**Figure 4 sensors-21-02123-f004:**
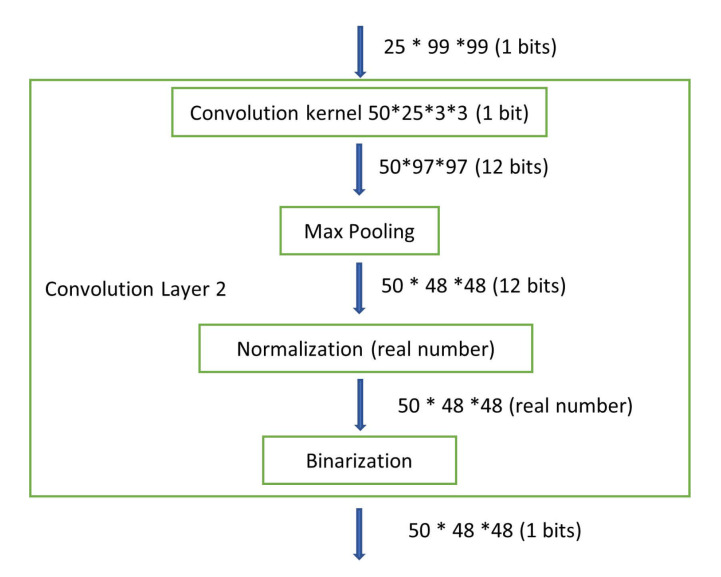
Describes the data type through the binarized neural network (BNN) convolutional layer 2.

**Figure 5 sensors-21-02123-f005:**
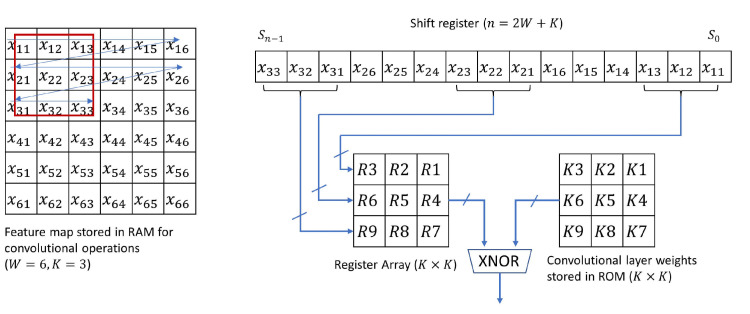
Shift register design.

**Figure 6 sensors-21-02123-f006:**
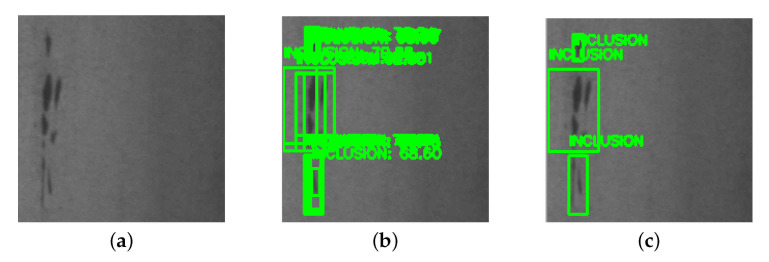
The results of non-maximum suppression algorithm: (**a**) describes the input image; (**b**) describes the defects localization results without non-maximum suppression; and (**c**) describes the defects localization results with non-maximum suppression.

**Figure 7 sensors-21-02123-f007:**
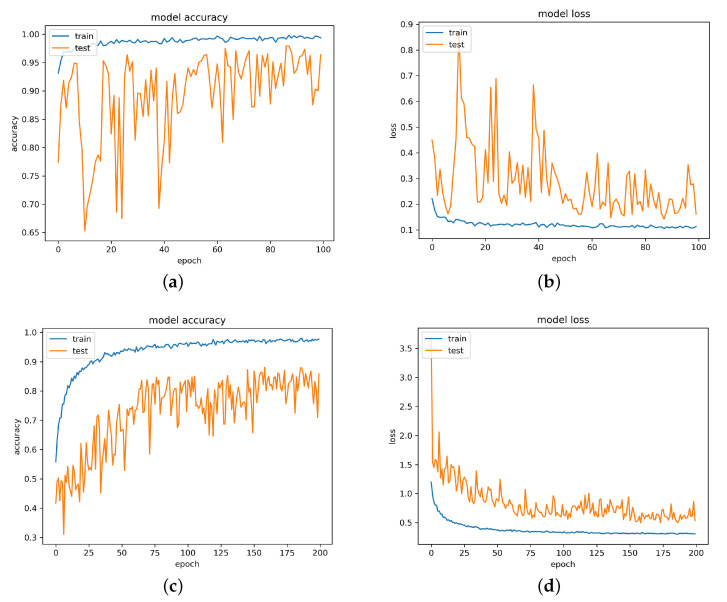
BNN model training results: (**a**,**b**) show BNN model accuracy and loss on defects identification (defective or defect-free); (**c**,**d**) show BNN model accuracy and loss on defect type classification.

**Figure 8 sensors-21-02123-f008:**
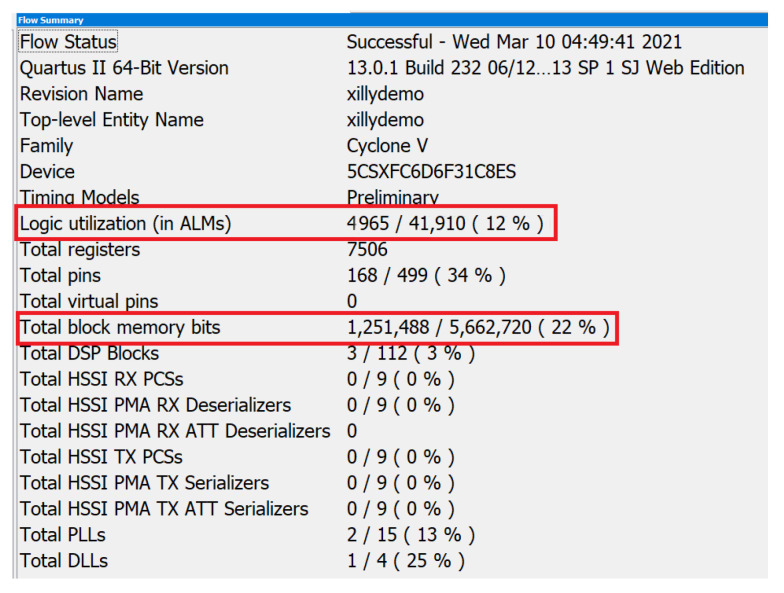
BNN model FPGA design synthesis results.

**Figure 9 sensors-21-02123-f009:**
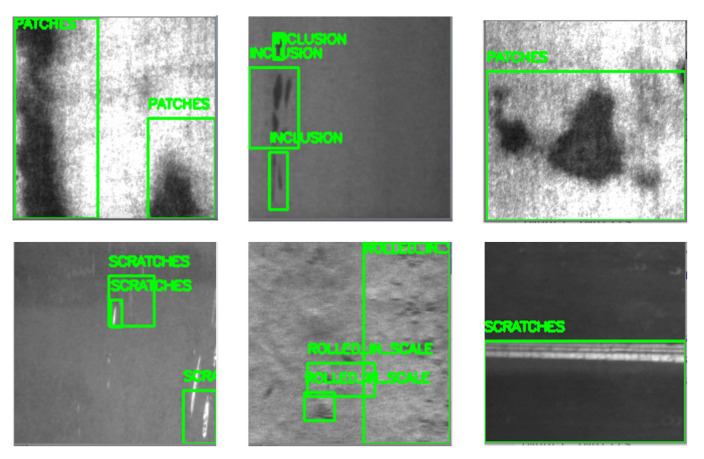
Defect inspection results.

**Table 1 sensors-21-02123-t001:** Describes different FPGA chips’ capability of replicating BNN modules.

FPGA Device	ALMs	Block Memory	Maximum No. of BNN Modules
Cyclone V 5CSXC6	41.5 K	5.6 Mb	4
Arria 10 GT1150	427.2 K	65.7 Mb	52
Stratix 10 GX10M	3466 K	308 Mb	246

## Data Availability

Not applicable.

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
