# Peer review of "FPGA-Based Acceleration on Additive Manufacturing Defects Inspection"

_sensors, 2021, doi:10.3390/s21062123_

Round 1

Reviewer 1 Report

FPGA are key in MEMS and a lot of devices, so the paper could be key for a journal. It must be improved by using several suggestions:

  • Conclusion: poor…please make a better effort in the line Considering the AM object production speed, the results can guarantee real-time quality inspection in the AM process. Besides, our FPGA-based BNN design can easily scale up by generating more BNN modules in a more powerful FPGA. See how massive use of acronyms…enhance the work to be defined. Terasic SoC Kit development board…..can anyone buy it??.

FPGA are in use in additive, but additive in polymers or in metals? They are quite different..in order to include some reference showing differences, a very recent one is A new hybrid process combining machining and selective laser melting to manufacture an advanced concept of conformal cooling channels for plastic injection molds, International journal of advanced manufacturing technology, total process control could be help by ANNs

Do you use a FPGA to run a Neural Network Architecture? Is it. I think this is a basic IoT system, good done. However additive must take into account that machine learning approach must take complex information, Smart optimization….boosting ensembles, Journal of manufacturing systems 48, 108-121, gave some ideas about how to complete the data set. A FPGA can be also programmed. A field-programmable gate array (FPGA) is an integrated circuit designed to be configured by a customer or a designer after manufacturing – hence the term "field-programmable". The FPGA configuration is generally specified using a hardware description language, so I see that a more complex model can be include…please discuss it.

Do you use a commercial FPGA, are others possible? Please discuss.

Paper is some short, a better method and writing is possible.

Reviewer 2 Report

Dear  Authors,

In my opinion, the manuscript titled FPGA-based Acceleration on Additive Manufacturing Defects Inspection   could be interesting for readers of the Sensors MDPI Journal, but in my opinion, it should be corrected and completed.Due to the listed below drawbacks my recommendation is "Reconsider after major revision".

Please refer to the comments/remarks listed below:

The article does not specify what its novelty is. It also doesn't follow from Introduction. Introduction is written inconsistently. In the presented solutions, together with accuracy, the calculation time and hardware are given, while for other examples not.

The article looks / feels like a research report. It contains a lot of detailed hardware information and a little scientific considerations. The novelty of the presented analyzes should be clearly stated

Moreover in my opinion, the linguistic style of the article should be improved.

The following statements are not used in scientific articles: 

  • In literature [15]...
  • The authors of [16]...
  • Paper [17]....

The caption in Fig. 3 is definitely too long and serves as a comment that should be included in the text of the article. 

Please justify the values of the constants in the formula (1).

In my opinion the notation 2*2, 100*100 or 25*99*99 is not elegant Please use 2x2, 100*100 or 25*99*99 notation instead.

The BNN model FPGA design synthesis results shown on Fig. 8 are hardly readable. Please prepare the table with the results.

Reviewer 3 Report

The paper proposes a real-time inspection, based on binarized neural network and FPGA, to identify different type of surface defects on products obtained by additive manufacturing (AM).

The study reports some interesting results, but there are many lacks to fill.

I suggest including the number of lines to make easier the subsequent reviewing process.

Abstract

  • Replace second with s;

Introduction

  • “Traditional defects inspection […] is limited.” – These phrases are questionable. Many inspections provide objective data on AM facilities. I suggest considering recent articles like:

https://ieeexplore.ieee.org/document/8932617

https://link.springer.com/article/10.1007%2Fs40684-017-0029-7

  • Replace milliseconds with ms.
  • Replace watts with W.
  • “The objective of […] by conclusions. – Put into evidence the paper innovation.

Research methods

  • Why did you choose to investigated 200*200 pixels images instead of other sizes?
  • “The modified dataset […] the ratio of 7:3.” – Clarify the meaning of the ratio 7:3.
  • “CNN models are […] vision techniques [20,21].” – Clarify the meaning of these lines.
  • “In our batch normalization layer, […] respectively.” – Define momentum and epsilon.
  • 200*200, 3*3 and 2*2 should have a unit of measurement.
  • Improve the resolution of Figure 2.
  • The comment in the caption of Figure 3 should be in the text and not there.
  • Improve the clarity of paragrapher 2.4 and its sub-paragrapher.
  • Provide some details about non-maximal suppression to help the reader in the image processing understanding.

Results evaluation

  • How were BNN model and defect classification accuracies evaluated?
  • Improve the resolution of Figure 6.
  • “The system on […] Processor.” – These sentences should be reported in Research methods section.
  • Check the caption of Table 1.

Round 2

Reviewer 2 Report

Dear Authors,

thank you for your comments.

All my remarks and doubts have been explained and commented on by the authors. The explanations are clear and satisfactory to me.

I am of the opinion that the article can be published in its current form. 

My recommendation is Accept in the present form.

Author Response

Dear Reviewer,

We are glad that you are satisfied with our previous responses.

We appreciate the time and effort that you dedicated to providing feedback on our manuscript and are grateful for the "Accept" recommendation. 

Reviewer 3 Report

The manuscript was improved, but some issues are not clear:

  • Lines 31-36 - My query n. 3 was not properly addressed. Suggested papers, which propose visual methods for investigating superficial defects in additive manufacturing, should be included and discussed.

https://ieeexplore.ieee.org/document/8932617

https://link.springer.com/article/10.1007%2Fs40684-017-0029-7

  • Lines 140-141 - This sentence does not partially reflect the reply for my query n.6. Moreover, how many mm does 1 pixel correspond to? This is extremely relevant for identifying the spatial resolution of the defect. Are there any references in literature that show 200 × 200 pixels as an acceptable size for such type of analysis?
  • Figure 6 – A scale in mm could be very useful to partially address to the previous query.

Round 3

Reviewer 3 Report

All my comments have been addressed. In fact, the authors have improved their article. In my opinion, the document can be considered for publication.